# Macromolecular Structure Assembly as a Novel Antibiotic Target

**DOI:** 10.3390/antibiotics11070937

**Published:** 2022-07-13

**Authors:** Scott Champney

**Affiliations:** Department of Biomedical Sciences, JH Quillen Collage of Medicine, East Tennessee State University, Johnson City, TN 37614, USA; champney@etsu.edu; Tel.: +(423)-928-1956

**Keywords:** penicillin-binding proteins, DNA polymerase III, RNA polymerase, ribosomes, crystallography, antisense oligonucleotides, tertiary and quaternary structures

## Abstract

This review discusses the inhibition of macromolecular structure formation as a novel and under-investigated drug target. The disruption of cell wall structures by penicillin-binding protein interactions is one potential target. Inhibition of DNA polymerase III assembly by novel drugs is a second target that should be investigated. RNA polymerase protein structural interactions are a third potential target. Finally, disruption of ribosomal subunit biogenesis represents a fourth important target that can be further investigated. Methods to examine these possibilities are discussed.

## 1. Introduction

There are very serious problems caused by the current increase in infectious diseases worldwide. Human infections by both microbial and viral sources are new threats to be contended with [1]. The COVID-19 global pandemic is one problem that fortunately is coming under control using highly effective vaccines and barrier strategies to slow the transmission of the virus. This is welcome news and the use of messenger RNA as a vaccine source is a novel and welcome technology [2]. However, a more serious and less controllable problem is the universal rise in multidrug-resistant microorganisms. This is a potentially greater problem that may ultimately prove more difficult to overcome. Resistance mechanisms include extracellular modification or destruction of antimicrobial agents [3]. Reduced import or enhanced efflux of drugs is another mechanism. Additionally, enzymatic target modification or mutational change to the target can also occur. The resistance problem is compounded by the ability of microbes to exchange resistance genes by the mechanism of horizontal gene transfer [4].

In some cases, few if any new antibiotics are available to combat multidrug-resistant microorganisms. Hospital-acquired infections by multidrug-resistant organisms are becoming a severe problem in many hospitals [5]. Current antimicrobial agents are increasingly less effective against a large number of infectious microorganisms [6]. Unfortunately, only a few novel drugs are currently being investigated [7] and fewer solutions to the resistance problem are being offered right now. Novel targets are currently being sought with some successes being reported [8].

One potential source of new targets for drug development is the inhibition of the assembly of essential macromolecular structures. Multi-protein targets with the potential for assembly disruption include the penicillin-binding proteins (PBPs), essential for cell wall construction. Two other important protein complexes are the bacterial DNA polymerase III and RNA polymerase macromolecular structures. Finally, the large 70S ribosome structure, essential for protein synthesis, is a fourth attractive target. Each of these structures can be investigated as a novel target for assembly inhibition. Our extensive investigations describing the inhibition of ribosomal subunit formation by ribosome-binding antibiotics are the stimulus for this proposition [9]. Macromolecular assembly inhibition by antimicrobials has been previously suggested by Bandyopadhyay [10].

Most current antimicrobial agents function by interfering with enzymatic activity essential for cell growth. For example, penicillin and related compounds inhibit the enzymatic functions of the penicillin-binding proteins needed for cell wall formation [11]. Fluoroquinolone antibiotics inhibit the topoisomerases needed for the enzymatic supercoiling of DNA required for replication to proceed [12]. Rifampicin and related antibiotics inhibit the RNA polymerase activity required for the initiation of transcription [13]. Ribosomal antibiotic targets are different [14]. The catalytic activity of the peptidyl-transferase structure in 23S rRNA is a target for some antibiotics such as chloramphenicol, which prevent amino acid polymerization. However, many other ribosome-targeting compounds inhibit the binding associations needed for ribosomal function in translation. For example, aminoglycoside drugs bind to the 16S rRNA in 30S subunits and stimulate mistranslation of mRNA. Macrolides and related antibiotics bind to 23S rRNA and block the egress of nascent peptides from the 50S exit tunnel. This distinction is important because almost all antimicrobial agents that target the ribosome bind to rRNA and not to ribosomal proteins. Macromolecular structure formation is different because tertiary and quaternary interactions are the targets, not catalytic functions. Interruption of the tertiary and quaternary interactions in each of these targets requires information about the exact three-dimensional structures of these four complexes [15]. For each of these targets, detailed structural information is available from X-ray crystallographic models and cryo-electron microscopy studies. Careful examination of the molecular interactions in these structures can identify potential target sequences for macromolecular complex disruption.

## 2. Penicillin-Binding Protein Structures as a New Antimicrobial Target

The PBPs are extracellular protein molecules which catalyze the cross-linking of the glycan chains in the bacterial cell wall. They cause the polymerization of the glycan strands through trans-glycosylation activity. They also generate cross-links between the glycan chains through trans-peptidase activity. This process generates the cell wall structure in most microorganisms [16]. There are variable numbers of high molecular mass proteins (HMMs) and low molecular mass proteins (LMMs) in different microbial species, ranging from three to eight distinct proteins. The tertiary structure of a number of different PBPs has been determined by X-ray crystallography [17]. These studies have revealed the similarities in the beta-lactam antibiotic binding sites for inhibition of cell wall biosynthesis. Disruption of the enzymatic formation of the peptidoglycan structure by these drugs results in the weakening of the cell wall structure and eventual death of the organism by osmotic pressure. Numerous penicillin and cephalosporin compounds inhibit the enzymatic activity of the PBPs and the mechanism of action of these drugs has been well investigated. The cephalosporins and carbapenems are additional families of antimicrobial agents that also impair the cross-linking activity of the substrates. Currently, new antimicrobial combinations are being developed, such as meropenem-vaborbactam and boronic acid derivatives. Resistance to these antimicrobials is typically caused by beta-lactamase proteins that impair the antibiotic function [11].

A potential novel target is the assembly of the PBPs into multi-protein quaternary complexes. Interactions between the various PBPs are being actively investigated based on the increasing numbers of available structures [18]. Quaternary interactions are not as well understood, but knowledge about the three-dimensional structures of many PBPs should reveal sites for protein–protein inhibitory possibilities.

## 3. DNA Polymerase Structure Formation as Another Antimicrobial Target

The DNA replication protein complex offers a second novel antimicrobial target. DNA polymerase III is the primary enzymatic structure in bacterial cells for DNA synthesis. The bacterial replication enzyme complex for DNA synthesis is a multi-protein structure consisting of ten protein subunits [19]. Its structure has been precisely determined by high resolution X-ray crystallography and cryo-electron microscopy [20]. The initiation of DNA replication is a very complicated process. It requires the identification of the replication origin sequences in DNA, the synthesis of an RNA primer sequence, the binding of the pol III complex to the origin and the extension of the primed DNA sequence in two directions. Nuclease removal of the RNA primer is an essential step as well. Accessory proteins such as the clamp-loader fasten the polymerase to the single stranded DNA to allow for processive extension of the new strands. A substantial amount of work has revealed the tertiary structures of the replication proteins and the quaternary interactions essential for forming the pol III holoenzyme. Fluoroquinolone antibiotics target the topoisomerase activity necessary for replication, but they do not interfere with the enzymatic polymerization process itself [12]. Searches for small molecule inhibitors of polymerization have been conducted with some success [21]. Stalling the assembly of the multi-protein polymerase III complex has not been well studied. This complexity allows for the possibility for intervention by small molecules and by competing antisense sequences.

## 4. RNA Polymerase Structure Assembly as a Novel Drug Target

The bacterial RNA polymerase is another multi-protein complex composed of a core set of proteins that interact with numerous axillary proteins. This machinery is a third potential target for disruptive agents. RNA polymerase formation involves the assembly of five essential proteins. Two alpha subunit molecules interact with beta and beta-prime proteins and with the omega protein to generate the core polymerase structure [22]. This quaternary complex is associated with a variety of sequence-specific sigma factor proteins that direct the core RNA polymerase to promoter sites in the DNA [23]. Numerous three-dimensional structures of RNA polymerase have been described by crystallographic analysis [13]. One well-investigated antibiotic inhibitor of RNA polymerase is rifampicin, which blocks elongation of the RNA sequence. A number of other different antimicrobials also inhibit the enzymatic activity of RNA polymerase [24]. These include four natural products that bind to the switch region of the polymerase. They have been identified as myxopyronin, corallopyronin, ripostatin and lipiarmycin. The assembly of the core polymerase is another potential novel target. Disruption of the quaternary protein interactions in this complex is another unique target possibility.

## 5. Ribosomal Subunit Assembly as a Fourth Antibiotic Target

A fourth target is the complex process of assembling the 70S bacterial ribosome, the machinery for protein biosynthesis in all cells. It is the largest macromolecular structure in microorganisms. Its assembly and functions in translation involve the association of numerous ribosomal proteins with specific rRNA sequences. The structures of the 70S ribosome and of its 50S and 30S subunits have been determined by X-ray crystallography by a number of different investigators [25,26]. This information has permitted investigations into the assembly process both in vitro and in vivo.

The formation of the 30S and 50S subunits of the ribosome involves the interactions of ribosomal proteins with the three ribosomal RNA molecules in a coordinated process that has been extensively investigated [27]. Ribosomal subunit biogenesis begins with the synthesis of a large precursor RNA molecule containing the 16S, 23S and 5S RNA sequences. This is cleaved into the three separate RNA molecules by ribonuclease III. Ribosomal protein synthesis is coordinated with RNA formation [28]. The basic steps in the assembly process are shown in Figure 1.

The larger 50S subunit is constructed by the association of 35 ribosomal proteins with both 23S and 5S ribosomal RNA species. The assembly pathway involves the formation of an intermediate 32S structure containing 23S and 5S ribosomal RNAs and about half of the subunit proteins. A conformational rearrangement allows the formation of a second intermediate, the 43S structure. This then adds the remaining proteins to generate the mature 50S subunit.

The assembly of the smaller 30S subunit involves a similar pathway. Immature 16S RNA interacts with a subset of proteins to form a 21S precursor structure. This undergoes a conformational rearrangement of the particle, permitting the addition of the remaining proteins to generate the final 30S particle. The assembly process is facilitated by the interaction with numerous chaperone proteins having temporary interactions with the maturing RNA structures [29]. Maturation of the structure also involves the reduction in size of both RNA species by specific ribonucleases [30].

The 70S ribosome is the target for a large number of subunit-specific antibiotics [31]. Different drugs affect specific functions of the 30S or 50S subunit in the translation process. For example, aminoglycosides bind to the 30S subunit and stimulate mistranslation of messenger RNA by impairing specific codon–anticodon recognition. The peptidyltransferase function of the 50S subunit is inhibited by chloramphenicol and streptogramin A antimicrobials, among others. Passage of the nascent peptide chain through the 50S exit tunnel is impaired by macrolides, ketolides, lincosamides, and streptogramin B compounds as other examples.

An important feature of most ribosomal-targeting drugs is their binding exclusively to ribosomal RNA and not to ribosomal proteins. This suggests that compounds binding to nascent RNA sequences in the subunit precursor particles should also impair subunit maturation [32]. Our extensive investigations have revealed the targeted disassembly of these macromolecular complexes by different ribosome-specific antibiotics [9].

We have shown that aminoglycoside antibiotic binding to 16S rRNA impairs subunit formation and results in the accumulation of the 21S intermediate [33,34]. This structure is degraded by cellular ribonucleases. A 32S precursor to the mature 50S subunit accumulates in cells treated with 50S subunit-specific antimicrobials [35,36]. Macrolide, ketolides, lincosamides and streptogramin B antibiotics promote the accumulation of the first 32S subunit precursor structure [37]. This assembly inhibition is illustrated in Figure 1. These studies were the stimulus that suggested that the biosynthesis of other macromolecular structures could also be impaired by novel compounds.

More than 30 proteins have been identified as chaperones necessary for ribosomal subunit biogenesis [29]. Many are required to facilitate the folding of the nascent ribosomal RNAs into their final tertiary structure. Others facilitate the acquisition of the appropriate three-dimensional sites for ribosomal protein binding. Protein–protein interactions are facilitated by the action of other novel chaperones. Some aid in the formation of mature secondary structures for nascent proteins emerging from the 50S subunit tunnel. Many of these required proteins are potential targets for antibiotic development. Interference with their essential functions in ribosomal subunit biogenesis represents a novel collection of potential drug targets.

Certain cellular ribonucleases are unique enzymes needed for the maturation of all three ribosomal RNA molecules [30]. Both the 5′ and 3′ ends of each RNA molecule are trimmed by specific RNase activities to generate the final mature molecules. These represent a collection of unique targets for antimicrobial development. Ribonuclease mutant strains of *E. coli* have an enhanced susceptibility to ribosomal antibiotics, demonstrating the essentiality of these enzymes [38,39]. Exploration of these types of compounds could lead to RNase target-specific antimicrobial agents.

## 6. Methods for the Identification of Antimicrobial Agents That Inhibit Macromolecular Target Formation

The tertiary and quaternary interactions between protein sequences in the macromolecular structures are likely to be of three types. The association may be by hydrogen bonding, electrostatic bonds or hydrophobic interactions [40]. Knowledge about these types of interactions can guide the search for methods to disrupt structure formation. Three types of approaches may be considered.

First, small molecule inhibitors may be found which can disrupt critical protein–protein interactions. Coupled with three-dimensional structure information from crystallographic analysis, this approach offers the potential for identifying novel drug-binding sites. Designer drugs may be synthesized, which can interrupt structure assembly based on the identified protein sequence interactions. Unlike antibiotics, which impair catalytic activity, these would be molecules that prevent critical protein–protein interactions instead [41,42].

Second, antisense oligonucleotides can be synthesized which associate with nascent mRNA sequences and impair the synthesis of proteins in the targeted structures. In addition, antisense oligonucleotides can be used to target rRNA sequences and thus impair translation or ribosomal subunit biosynthesis [43].

Third, genome sequence searches are another avenue for antimicrobial development. Synthetic oligonucleotide sequences can be used to stimulate the overproduction of essential peptide sequences that would compete with the interacting proteins in the macromolecular structures. These can potentially interfere with either tertiary or quaternary structure formation [44]. Artificial intelligence genome sequence analysis has identified binding sequences for novel compounds with unique inhibitory activities.

The inhibitory activity of novel antimicrobial agents can be amplified by the use of drugs in combination [45]. Two small-molecule inhibitors could be used in combination to simultaneously inhibit an essential enzymatic activity and also interfere with macromolecular interactions. For example, azithromycin inhibition of protein synthesis and ribosome assembly in *Staphylococcus aureus* was significantly enhanced by the inclusion of rifampicin or ciprofloxacin [46]. Other drug combinations have shown similar stimulatory effects. Another interesting approach is the use of adjuvant molecules to amplify the inhibitory effects of certain antimicrobial drugs [47]. Three main types of antibiotic adjuvants have been developed, which include β-lactamase inhibitors, efflux pump inhibitors and outer membrane-disruption agents. Microorganism-specific bacteriophage infection is another unusual adjuvant approach.

This proposal represents an attempt to apply some of the information gained from our studies on antibiotic inhibition of bacterial ribosome assembly to other macromolecular structures. Importantly, knowledge of the three-dimensional structures of the four potential targets permits identification of critical interactions which can be targeted. Novel approaches such as these are necessary to attack the increasingly serious problem of antimicrobial resistance.

## Figures and Tables

**Figure 1 antibiotics-11-00937-f001:**
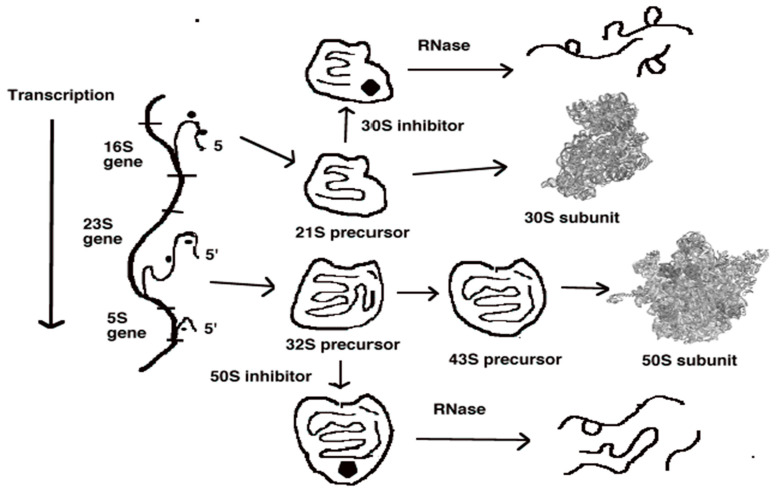
A model for ribosomal subunit assembly and inhibition by antibiotics. 16S rRNA transcription followed by ribosomal protein addition gives a 21S precursor particle, which adds additional proteins to yield 30S subunits. Transcription of 23S and 5S rRNA is followed by the formation of both 32S and 43S intermediates, which lead to 50S subunit formation after specific ribosomal protein addition. In the presence of antibiotics, assembly stalls at the first defined intermediate particle, which can bind the appropriate antibiotic. Ribonucleases degrade the stalled intermediate particle, reducing net subunit formation. Note that 16S rRNA transcription and 30S assembly precede 23S and 5S rRNA transcription and 50S assembly [9].

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
