# Peer review of "Macromolecular Structure Assembly as a Novel Antibiotic Target"

_antibiotics, 2022, doi:10.3390/antibiotics11070937_

Round 1

Reviewer 1 Report

Macromolecular interaction could be the one of the most probable target for novel class of antibiotics to combat with antibiotic resistance.  But scope is too broad and shallow. If all four aspects of macromolecular interaction might be broken down to each subject and discussed in more detail such as with more emphasis on the structural aspects or biochemical aspects or else, it could be more appropriate to be published in antibiotics. Or if the reason why inhibition of macromolecular interaction could not bring the success  to the present might be discussed, it could be more interesting. 

Author Response

I appreciate the reviewer’s comments. This review was not intended to be a comphrensive article discussing macromolecular targets in detail. My purpose was to suggest that these four examples could be used as a way to stimulate new thinking about antimicrobial targets. My research on the antibiotic inhibition of ribosomal subunit biogenesis was the basis for this proposal.

Reviewer 2 Report

Mr. Champney discusses inhibition of macromolecular structure formation as a novel and insufficiently explored drug target. The article is fluid and easy to understand, although I do not understand why it is classified as a hypothesis. However, there are some points that should be addressed.

According to the instructions for authors (direct citation), reference numbers should be placed in square brackets [ ] and precede punctuation; for example, [1], [1-3], or [1,3]. For embedded citations in text with pagination, use both brackets and parentheses to indicate the reference number and page numbers; for example, [5] (p. 10). or [6] (pp. 101-105).

All references are in round brackets and should be corrected.

Some references are in bold, some references have bold brackets (one or both), for example in lines32, 36, 45, 49, 103, 115,187, 190. these should be corrected.

Line 19: According to Merriam-Webster, Covid19 is not the correct spelling. You should write COVID -19 or more rarely Covid-19 or covid-19.

Lines 24 and 25: Add reference 3 to the end of the sentence in line 27.

The quality of Figure 1 is not acceptable for an article. It looks like it was drawn in MS Paint and looks really bad. The author should use one of the paint programs, such as Inkscape or maybe even ChemDraw, which also has some drawing features.

Line 172, I think there is too much space before the word Macrolide.

Line 222, certain antimicrobial drugs. instead of certain antimicrobial drugs,

Line 223, β-lactamase instead of b-lactamase.

The author should check the references carefully because there are many errors in them. For example. In line 235 a comma is missing, in line 241 the page numbers are written in italics, on page 248 doi is missing, etc.

Author Response

Reviewer 2.

I appreciate the reviewer’s comments. All of the suggested formatting issues have been addressed. I prefer to leave the figure in black and white to illustrate the effects of antimicrobials on ribosome biogenesis in a general way. 

Reviewer 3 Report

This is hypothesis based on the current knowledge of target structure for antibiotics. I have some comments to give the author to improve the manuscript as the following:

1. Introduction: Line 32: Please change from "Nosocomial" to "Hospital-acquired"

2. Separate to sub-heading sections for each macromolecule/antibiotic-target molecule will be enhance the reader to read easily and figure out the author message.

In an example, Cell-wall target; Protein-synthesis target; DNA synthesis target or other, etc.

3. Example of novel/currently antibiotics should be described such as meropenem-vaborbactam, imipenem-relebactam

Author Response

Reply to Academic Editor Each of the points raised by the reviewer have been addressed.
1. Nosocomial has been changed to hospital-acquired.
2. Five labeled subheadings have been added to the paper.
3. Novel current antimicrobials have been added to several of the sections including meropenem-vaborbactam, boronic acid, myxopyronin ,corallopyronin, ripostatin and lipiarmycin and linezolid.